# Bone Marrow Mesenchymal Stromal Cells in Multiple Myeloma: Their Role as Active Contributors to Myeloma Progression

**DOI:** 10.3390/cancers13112542

**Published:** 2021-05-22

**Authors:** Patricia Maiso, Pedro Mogollón, Enrique M. Ocio, Mercedes Garayoa

**Affiliations:** 1University Hospital Marqués de Valdecilla (IDIVAL), University of Cantabria, 39008 Santander, Spain; 2Cancer Research Center (IBMCC-CSIC-USAL), University Hospital of Salamanca (IBSAL), 37007 Salamanca, Spain; pmog@usal.es (P.M.); mgarayoa@usal.es (M.G.)

**Keywords:** multiple myeloma, bone marrow mesenchymal stromal cells, myeloma progression

## Abstract

**Simple Summary:**

Multiple myeloma is a cancer of immunoglobulin-secreting cells that accumulate in the bone marrow. Mesenchymal stromal cells are important components of the bone marrow microenvironment interacting with myeloma cells and having a pivotal role in the progression of the disease. Here we first review studies that have highlighted structural and functional differences between mesenchymal stromal cells derived from healthy donors and myeloma patients, and propose a model for the transition from the normal to the myeloma-condition of these cells. Next, we underscore the contribution of mesenchymal stromal cells to the promotion of myeloma growth and survival, development of drug resistance, dissemination and homing, myeloma bone disease, and the establishment of a pro-inflammatory and immunosuppressive microenvironment. It appears as if as a result of myeloma-mesenchymal stromal cell cross-talk, mesenchymal stromal cells in myeloma patients have converted into active contributors to the pathophysiology of the disease.

**Abstract:**

Multiple myeloma (MM) is a hematological malignancy of plasma cells that proliferate and accumulate within the bone marrow (BM). Work from many groups has made evident that the complex microenvironment of the BM plays a crucial role in myeloma progression and response to therapeutic agents. Within the cellular components of the BM, we will specifically focus on mesenchymal stromal cells (MSCs), which are known to interact with myeloma cells and the other components of the BM through cell to cell, soluble factors and, as more recently evidenced, through extracellular vesicles. Multiple structural and functional abnormalities have been found when characterizing MSCs derived from myeloma patients (MM-MSCs) and comparing them to those from healthy donors (HD-MSCs). Other studies have identified differences in genomic, mRNA, microRNA, histone modification, and DNA methylation profiles. We discuss these distinctive features shaping MM-MSCs and propose a model for the transition from HD-MSCs to MM-MSCs as a consequence of the interaction with myeloma cells. Finally, we review the contribution of MM-MSCs to several aspects of myeloma pathology, specifically to myeloma growth and survival, drug resistance, dissemination and homing, myeloma bone disease, and the induction of a pro-inflammatory and immunosuppressive microenvironment.

## 1. Multiple Myeloma and the Bone Marrow Microenvironment

Multiple Myeloma (MM) is a B cell neoplasm characterized by the bone marrow infiltration by clonal plasma cells secreting pathological immunoglobulins that can be detected in serum or urine. It represents a paradigm of a disease in which the progress in understanding the biology of the malignancy has resulted in prognostic and therapeutic advances, which have led to an improvement in patients’ outcomes [1]. First, MM is a model of transformation from premalignant asymptomatic conditions (monoclonal gammopathy of uncertain significance -MGUS- and smouldering myeloma -smMM-) into active stages when disease-associated symptoms such as anemia, hypercalcemia, renal impairment, or bone lesions may appear. The study of this evolution has led to the clinical evaluation of the concept of early intervention for these asymptomatic patients [2]. Prognosis is another field of research in MM that started with the use of clinical parameters such as albumin or β2 microglobulin, continued with the discovery of molecular and cytogenetic abnormalities conferring adverse prognosis, and more recently, dynamic markers like the evaluation of the measurable residual disease (MRD) have been incorporated to guide treatment decisions. However, the most significant advances have been developed in the therapeutic area, with more than 10 novel agents approved since the early 2000s. As mentioned, this progress has derived from a better knowledge of the disease biology that prompted the evaluation of new molecules targeting mechanisms essential for tumor cell survival. Moreover, in the last years, we have seen an explosion of immunotherapeutic strategies with the appearance of monoclonal antibodies, bispecific T engagers, or even CAR-Ts that have revolutionized treatment expectations for this disease.

Although many pathogenic factors are tumor cell-autonomous in cancer, they are usually insufficient to induce progression and metastasis, and a permissive or even favouring microenvironment is required for frank malignancy to emerge and later to evade apoptosis. Today, it is widely accepted that the tumor microenvironment plays an active and pivotal role in acquiring the so-called tumor hallmarks, enabling tumor growth and evasion of apoptosis, the development of drug resistance, and metastasis [3,4]. 

MM cells accumulate within the bone marrow (BM) niche, which consists of a complex microenvironment of cellular components including mesenchymal stromal cells (MSCs), fibroblasts, adipocytes, endothelial cells (ECs), osteoclasts (OCs), osteoblasts (OBs), immune cells and hematopoietic cells, together with a non-cellular compartment of the extracellular matrix and the liquid milieu of cytokines, growth factors, chemokines and extracellular vesicles [5,6,7]. Today it is well established that besides genetic and epigenetic alterations occurring in myeloma cells, the BM microenvironment plays a pivotal role in mediating survival, proliferation, drug resistance, and progression of the disease [6,8,9,10,11]. MM is a prototype of malignancy characterized by complex bi-directional interactions between tumor cells and the BM microenvironment. This review will focus on the interactions of myeloma cells and mesenchymal stromal cells (MSCs), and how these interactions contribute to the transition from “normal MSCs” to “myelomatous MSCs”. Finally, we will go through the reported contribution of MSCs to several aspects of MM pathology, i.e., growth and survival of myeloma cells, dissemination and homing, myeloma bone disease, the induction of a pro-inflammatory microenvironment, and the development of drug resistance. 

## 2. Characterization of Mesenchymal Stromal Cells in Multiple Myeloma

### 2.1. Mesenchymal Stromal Cells (MSCs)

Mesenchymal stromal cells (MSCs) were originally described by Friendenstein et al. more than 50 years ago, as cells from the BM with fibroblast-like appearance, capable of differentiation into osteocytes, chondrocytes, adipocytes, tenocytes, and myocytes [12]. These adult multipotent cells comprise a heterogeneous population, which was later shown also to differentiate into cells from the other germ layers (including neurons, cardiomyocytes, and hepatocytes). Besides the BM, MSCs have been isolated from a wide range of adult tissues, such as adipose tissue, liver, skeletal muscle, placenta, or lung (see review [13]). Their ease of isolation and expansion, together with their capacity to home towards injured tissue and multi-lineage and immunomodulatory potentials, have granted their clinical use in regenerative medicine and the treatment of immune disorders [13,14].

However, the lack of a specific cell surface marker for this cell population prompted the International Society of Cellular Therapy (ISCT) to define minimum criteria to allow comparison of MSCs between different studies: adherence to plastic under standard culture conditions, positivity for CD105, CD73 and CD90 (≥95%) and negativity for hematopoietic cell surface markers (CD34, CD45, CD14/CD11b, CD79a/CD19 and HLA-DR (≤2%)), and capacity for in vitro differentiation into osteocytes, adipocytes, and chondrocytes under appropriate stimuli [15]. Other standardized assays have been proposed for the use of MSCs for clinical purposes [16,17].

### 2.2. Physiological and Pathological Roles of MSCs in the BM

Despite being at a very low proportion in the BM (only 0.01 to 0.001% of mononuclear cells) [18], under physiological conditions, MSCs support the maintenance and differentiation of hematopoietic lineages, regulate bone homeostasis and contribute to the spatial delimitation of cellular niches. However, in MM, and as part of the BM microenvironment, MSCs play a crucial role in different aspects of the pathology of the disease, which will be discussed in Section 3. These pathological functions are mediated by a complex intercellular cross-talk of myeloma cells and MSCs, other cells of the BM microenvironment, and with the extracellular matrix. In this sense, myeloma to MSC interactions through adhesion molecules are prominent: VLA-4 (Very Late Antigen 4; α4β1) to VCAM (Vascular Adhesion Molecule); LFA-1 (Leukocyte Function associated Antigen 1) and MUC-1 (Mucin 1, cell surface-associated) to ICAM (Intercellular Adhesion Molecule); AXIIR (Annexin II Receptor) to AXII (Annexin II); Notch1/2 and Jagged1/2 to Jagged1/Dll (Delta-like) and Notch 1 [5,19,20,21]. In addition, a great body of soluble factors (cytokines, chemokines, growth, and differentiation factors) has been broadly shown to mediate these reciprocal interactions. More recently, also membrane-limited extracellular vesicles (EVs), both exosomes (Ø 50–150 nm) and microvesicles (Ø 150 nm–1 µm), which in MM are essentially produced by myeloma cells and MSCs, mediate the horizontal transfer of protein, lipids, and nucleic acids in their cargo, conveying information either locally or to cells at distant sites of the BM [22,23]. In fact, EVs have already been described to play roles in myeloma proliferation, development of osteolytic lesions, drug resistance, angiogenesis, and progression (see reviews [24,25]).

### 2.3. Comparison of MSCs from Healthy Donors (HD-MSCs) and Myeloma Patients MM-MSCs)

A considerable body of knowledge has been accrued comparing human MSCs from the BM of healthy donors (HD-MSCs) and those derived from myeloma patients (MM-MSCs) using different experimental settings and techniques. This next section will review selected in vitro studies in this regard, differentiating data reported from expanded/freshly isolated MSCs, MSCs in monoculture or co-culture with myeloma cells, and MSCs from 2D/3D in vitro cultures. In this way, we will try to discern the molecular processes underlying the transition from HD-MSCs to MM-MSCs.

#### 2.3.1. Studies from MSCs after Expansion vs. Fresh MSCs

Due to the low proportion of MSCs in BM aspirates, most studies have been conducted after isolation of MSCs by adherence to plastic and subsequent in vitro expansion for a low number of passages (from 2 to 4). Besides, culturing allowed the progressive loss of other adherent cell types in the sample and increased MSC purity. These studies enabled the assessment of functional and phenotypic properties found different in MM-MSCs and HD-MSCs. As compared to HD-MSCs, and despite some discrepancies between studies, MM-MSCs showed reduced osteogenic potential as demonstrated by reduced matrix mineralization and alkaline phosphatase (ALP) expression and activity [26,27,28], together with diminished expression of bone formation markers and transcription factors involved in osteogenic differentiation [29]. MM-MSCs also presented a reduced proliferation rate [30], increased angiogenic potential and secretion of angiogenic factors [31], and reduced efficiency to suppress T-cell proliferation [26,32,33]. The expression and secretion of growth factors, cytokines, chemokines, and factors negatively affecting osteogenic function was higher in MM-MSCs than in HD-MSCs (IL-1β IL-3, IL-6, IL-10, BAFF, GDF15, TNFα, TGF1β, DKK1, RANKL, and AREG) [26,27,33,34]. In addition, MM-MSCs showed an early senescence state with increased cell size and accumulation of cells in S phase, and a characteristic senescence-associated secretory profile (higher expression of HGF, IGF-2, IL-6, IL-8, MCP-1, MIP-1a, DKK1, and VEGF in MM-MSCs) [26]. In gene expression profiling studies, among the differentially expressed genes between MM-MSCs and HD-MSCs, a large proportion of genes were involved in tumor-microenvironmental cross-talk, and were implicated in tumor-support (e.g., IL-6, IL-1β, AREG, GDF15), angiogenesis (e.g., ANGPTL4, PAI-1), and contribution to bone disease (e.g., NPR3, WISP1). In another study, a differential transcriptional pattern was found between MM-MSCs, but not in OBs, when associating the occurrence of bone lesions within MM patients [35]. 

In search of potential alterations that could be causing the above functional and phenotypic differences of MM-MSCs, we conducted an array-based comparative genomic hybridization analysis. Only MM-MSCs bore several non-recurrent chromosomal gains and losses (>1 Mb) as well as discrete (<1 Mb) genomic alterations [36]. Although the significance of those genomic imbalances remains to be determined, their non-recurrent nature excluded them as being responsible for the functional and gene expression differences found in MM-MSCs. Nevertheless, the fact that those genomic alterations were not found in HD-MSCs maintained in the same culture conditions, at least may reflect a predisposition of MM-MSCs for genomic instability because of previous exposure to MM cells. Other studies have also identified genomic aberrations in patient-derived MSCs from MM and other hematological malignancies [37,38,39]. 

MicroRNAs (miRNAs) are endogenous 22-nucleotide long non-coding RNAs that regulate gene expression by binding to the 3´ untranslated region (UTR) of their target mRNAs and promoting mRNA degradation and/or inhibiting RNA translation. MiRNAs have been shown to play an important role in the osteogenic lineage commitment of MSCs by targeting the expression of transcription factors and components of the primary bone formation pathways [40]. Important differences in the expression of specific miRNAs have been found for HD-MSCs and MM-MSCs when subjected to osteogenic differentiation conditions. Xu et al. observed that miR-135b was upregulated in MM-MSCs and impaired their osteogenic differentiation by targeting SMAD5 [41]. Similarly, miR-138 was significatively increased in MM-MSCs compared to HD-MSCs [42], and inhibition of miR-138 resulted in enhanced osteogenic differentiation of MM-MSCs, being ROCK2, TRPS1, and SULF2 potential miR-138 targets. The expression of other miRNAs has been found upregulated in MM-MSCs (miR-221, miR203a-3p.1, miR-223) or decreased (miR-342, miR-363, or miR-29b) to that of HD-MSCs, modulating the expression of genes involved in osteoblastogenesis (see Raimondi et al. [43,44] for reviews). 

Conversely, other studies used single-cell RNA sequencing optimized for low-cell numbers of flow cytometry-sorted BM MSCs to transcriptionally characterize samples from both origins without in vitro expansion [45]. These studies showed that MSC frequency was significantly higher in patients with active myeloma, suggesting a differentiation blockade responsible for accumulating MSCs to support a protective microenvironment for myeloma cells. Principal component analysis of gene expression data also showed that in vitro expansion may magnify the differences between HD- and MM-MSCs. Additionally, MM-MSC transcriptional signature was found enriched in functions related to MM pathogenesis, such as IL-17 pathway and TNF signaling via NF-κB, osteoblastogenesis inhibition, MSC proliferation, immune-suppressive potential, and a reinforced pro-adipogenous phenotype (upregulation of Peroxisome Proliferator-Activated Receptor (PPAR) signaling and genes for cholesterol efflux and lipid metabolism). 

In another series of studies, Schinke et al. [46] established a 34 MSC-specific gene expression signature capable of distinguishing the transcriptional profile of MSCs from MM, MGUS, smMM, treated myeloma patients at complete response and heathy controls in BM biopsy samples. These data underscored that the expression of MSC genes within the BM microenvironment varies substantially between different disease stages. Moreover, a prognostic gene score based on 3-MSC specific genes (COL4A1, NPR3, and ITGBL1) was able to predict progression-free survival in MM patients and progression from smMM into MM, thus highlighting the contribution of the surrounding microenvironment in the progression of the disease.

#### 2.3.2. Studies of MSCs in Monoculture vs. MSCs in Co-Culture with MM Cells

Information about HD- or MM-MSC characteristics has also come from many studies in which expanded MSCs have been exposed to myeloma cells, reminiscent of their BM situation. Enhanced production of IL-6, IL-10, TNF-α, OPN, and especially of HGF and BAFF was observed by MM-MSCs after exposure to the RPMI8226 cell line [34]. Of interest, the authors noted a continued production of cytokines by MM-MSCs after several weeks of in vitro cultivation in absence of MM cells, suggesting the presence of an autocrine stimulation pathway for cytokines for MM-MSCs, and the possibility that this may promote relapse even during disease remission. Regarding the early senescence status observed in MM-MSCs after in vitro expansion, various studies support the MM-cell induced senescence as a contributing factor to the altered phenotypic characteristics of MM-MSCs. Kanehira et al. [47] found that co-culture of the IM9 myeloma cell line with MM-MSCs augmented the lysophosphatidic acid (LPA) signaling in MSCs. Since the expression of LPA receptor 1 (LPA1) is higher in MM-MSCs than in HD-MSCs, signaling through LPA1 (and not through LPA3) determined the induction of a pro-senescence profile in MM-MSCs and their transdifferentiation into tumor-associated fibroblasts, which promoted MM progression and tumor angiogenesis in in vivo models [47]. Besides, co-culture of myeloma cells from patients or myeloma cell lines with MSCs decreased the expression of Dicer1 and that of miR-93/miR-20a in the latter, which was associated with elevated expression of the cell cycle inhibitor p21. This resulted in senescence of MSCs, reduced osteogenic and increased adipogenic differentiation, and promotion of MM cell growth [48]. 

Other authors have analyzed gene expression changes induced in MSCs after interaction with myeloma cells. Our group studied the transcriptomic profile induced in HD-MSCs and MM-MSCs after direct transwell co-culture with the MM.1S cell line [49]. We found “commonly” deregulated genes in HD- and MM-MSCs functionally involved in tumor microenvironment cross-talk, myeloma growth induction and drug resistance, angiogenesis, and signals for OC activation and OB inhibition, which were suggested to reflect changes occurring in MSCs at initial phases of myeloma disease. CXCL1, CXCL5, CXCL6, and IL-8 were among the most highly deregulated genes in this subset. In contrast, other genes induced by co-culture were exclusively deregulated in MM-MSCs, and functional signatures linked those genes to RNA processing and splicing, activation of the ubiquitin-proteasome pathway, cell cycle regulation, cellular stress, and non-canonical Wnt signaling. Following the same line of reasoning, exclusively deregulated genes in MM-MSCs were suggested to represent expression changes of MSCs at more advanced stages of the disease, being Neuregulin3 and Norrie Disease Protein functionally validated genes within this second group. Another study also analyzed the transcriptomic signature of both HD- and MM-MSCs and pre-OBs after 24 h of direct contact with the INA-6 cell line [50]. Differential expression of genes in MSCs was related to disease progression (plasma cell (PC) homing, adhesion, enhanced angiogenesis) and tumor-induced bone loss (OC-derived coupling factors, increased adipogenesis, and inhibition of OB differentiation). One of the most upregulated genes in MSCs after myeloma contact was angiopoietin-like 4 (ANGPTL4), which mediates multiple roles in myeloma cell attachment, angiogenesis, regulation of lipid metabolism, and OC resorption [50,51]. Overall, these transcriptomic studies underscored the multiple deregulated genes in MSCs after interaction with myeloma cells, which partially resemble the phenotypic and functional differences observed between HD- and MM-MSCs in monoculture. Furthermore, they highlight the pivotal role of MSCs in supporting MM growth, angiogenesis, impaired OB differentiation, pro-adipogenesis, and OC formation in relation to the pathophysiology of the disease. 

MiRNAs have been recognized as essential performers in myeloma cells’ interactions with the BM microenvironment, and specifically with MSCs [52,53]. Interaction of myeloma cells with MSCs through direct contact and soluble factors has been shown to modify miRNA expression in both cell types, which in the case of MSCs may affect the expression of genes involved in suppression of OB differentiation but also tumor-promotion [43]. Thus, although increased levels of miR-135b, miR-138, and miR-21 were observed in MM-MSCs as compared to HD-MSCs, adhesion of myeloma cell lines to HD-MSCs further increased their levels, indicating the involvement of MM induction in the reduced osteogenic potential of MSCs [41,42,54]. As previously commented, the senescence-secretory profile induced in MM-MSCs because of interaction with myeloma cells was linked to decreased levels of miR-93/miR-20a, which in turn associated with elevated expression of the cell cycle inhibitor p21 [48]. Furthermore, the transfer of miRNAs and other ncRNAs from myeloma-derived EVs to MSCs has been reported to regulate gene expression and functions of MM-MSCs (e.g., increasing their cytokine secretion and proliferation, inducing their transformation to cancer-associated fibroblasts, and negatively regulating OB differentiation) [55,56,57]. Similarly, ncRNAs are also conveyed from MM-MSC-derived EVs into myeloma cells. The role of EVs targeting MSCs or derived from MM-MSCs in MM pathophysiology will be reviewed in the next section of this manuscript; for the roles of EVs on MSCs in MM, also see comprehensive reviews [24,58]. 

Studies from Adamik et al. showed that interaction of MM cells with MSCs induced the binding of the transcriptional repressor Gfi1 (growth factor independence-1) to the Runx2 promoter together with the chromatin modifier Enhancer of Zeste homolog 2 (EZH2), histone deacetylase 1 (HDAC1), and Lysine-specific demethylase 1 (LSD1) [20,59]. These chromatin modifiers deposit repressive chromatin marks and epigenetically block Runx2 transcription and osteogenesis, while at the same time contribute to the pathologic switch of MSC differentiation towards adipogenesis. Interestingly, the Runx2 epigenetic repression occurred 36–48 h after exposure of MM cells to MSCs. Moreover, Runx2 chromatin repressive marks were maintained four days after MM cell removal and were present in isolated and in vitro expanded MM-MSCs [60]. Akin to the latter studies, widespread DNA methylation alterations of BM isolated and expanded MM-MSCs from different myeloma stages compared to HD-MSCs have been found [61]. In particular, methylation alterations in Homeobox genes and other genes involved in osteogenic differentiation were shown to associate with altered expression along with myeloma progression. Of note, MM-MSC DNA methylation changes could be partially recapitulated by exposure of HD-MSCs to myeloma cells. Overall, these data suggest the involvement of both chromatin remodeling- and DNA methylation-based epigenetic mechanisms in the maintenance of suppressed OB differentiation of MM-MSCs after cultured in vitro in the absence of myeloma cells, and perhaps also in the persistence of unhealed MM bone lesions even after remission of active disease [60,61,62]. Since DNA methylation alterations in MM-MSCs are observed across the whole genome [61], it can be envisioned that not only a suppressed OB function but also the observed tumor-promoting features of MM-MSCs after expansion may rely on heritable epigenetic marks previously established by MSC-MM interactions in the BM [63].

#### 2.3.3. Studies of MSCs in 2D vs. 3D In Vitro Platforms 

Without any doubt, in vitro models that better mimic the physiologically relevant three-dimensional nature of the BM microenvironment, including adhesive, mechanical, and chemical cues from cells and the extracellular environment [64], may render better platforms to elucidate the MM and MSC interactions. When the cytokine production of MM-MSCs in response to the RPMI8226 myeloma cell line was examined in a 3D culture of gelatine sponge scaffolds, higher secretion of IL-11 and HGF and less IL-10 was observed as compared to 2D cultures [65]. Reagan et al. [66] established a 3D tissue-engineered bone (TE-bone) model based on silk scaffolds with a mineralized bone matrix, which was used to recapitulate the in vivo interactions of myeloma, MSCs, and endothelial cells. Specifically, this model was used to identify miRNAs mediating the myeloma-induced dysfunctional osteogenesis on MSCs. Among the most deregulated miRNAs in MSCs, downregulated miR-199a-5p was shown to have anti-osteogenic effects, and was also found downregulated in MM-MSCs. In other study series, MM-MSC-derived exosomes were shown to have differential miRNA and proteomic profiles compared to those from HD-MSCs. When MM-MSCs-exosomes were administered in the TE-bone model, increased myeloma growth was observed, whereas MM growth was even reduced when TE-bones were loaded with HD-MSCs-exosomes [67].

Myeloma BM co-cultures in gelatin scaffolds using the 3D rotatory culture bioreactor technology [68] also engaged functional myeloma-stroma interactions. Consistently, pro-survival signaling, cell adhesion, and soluble factor-mediated drug resistance were shown to be significantly higher in 3D than in 2D parallel co-cultures. Other 3D models, based on hydrogel [69] or fibrinogen gel cultures [70] of myeloma cells, endothelial cells, and MM-MSCs have recapitulated various aspects of the BM niche and their interactions and proven suitable for prediction of therapeutic response of MM patients ex vivo to various classes of drugs. 

### 2.4. Transition from HD-MSCs to MM-MSCs

Overall, the above-exposed data strongly suggest that multiple mechanisms in the BM govern the evolution from HD-MSCs to MM-MSCs. We propose (see Figure 1) that the interactions of HD-MSCs with myeloma cells as mediated by cell adhesion, soluble factors, and through extracellular vesicles are key and initiating elements of this transition, and perhaps for progression from MGUS/smMM to symptomatic stages of the disease. These interactions would progressively mediate dysregulation of gene expression, changes in the miRNA and epigenetic profiles, and perhaps even genomic abnormalities which would shape functional and phenotypic changes from HD-MSCs to MM-MSCs. Transversal factors affecting these changes in MSCs would be hypoxic conditions due to myeloma growth, the featured immunosuppressed microenvironment of MM, and patient aging.

Besides, and as suggested from various studies, inherited epigenetic modifications (expression of miRNAs, chromatin marks, and DNA methylation) established by myeloma-MSC interactions in the BM, could be at least partially responsible for the persistence of differences between HD- and MM-MSCs after a long-term absence of interactions with myeloma cells, such as those observed after in vitro expansion of MSCs. In addition, these epigenetic modifications may mediate a supportive role of MSCs to residual tumor cells in the BM and contribute to myeloma relapse. These considerations also raise the concept that the deregulation of epigenetic modifiers seems to be critical in the establishment, maintenance, and progression of pathological MM-MSC interactions in MM. However, since epigenetic mechanisms are reversible, the therapeutic use of epigenetic inhibitors, or even epigenetic inhibitors in combination with other anti-myeloma agents, offers at the same time new possibilities for the treatment of MM [63].

## 3. Biological Roles of MSCs in MM pathology

### 3.1. Contribution of MM-MSCs to Tumor Growth and Survival

MM tumor cells grow predominantly in the BM, and the cellular and non-cellular components of the MM BM microenvironment play an essential role in supporting MM cell proliferation, survival, migration, and chemoresistance [9]. Specifically, myeloma cells interact with MSCs, and far from being a passive relationship, mutual modulation of phenotype, proteome, and function is observed as a consequence of this cross-talk [71].

In the BM, MM cells adhere to MSCs and extracellular matrix (ECM) proteins through adhesion molecules. MM cells bind to type I collagen, fibronectin, and hyaluronan in the ECM via syndecan 1 (CD138), VLA-4, and CD44 respectively. VLA-4, LFA-1, MUC-1, or CD40 present on MM cells bind to VCAM-1, ICAM-1, or CD40L on MSCs. Adhesion of tumor cells to MSCs activates many pathways resulting in an induction of cell cycle progression and anti-apoptotic proteins, and inhibition of pro-apoptotic signaling pathways in MM cells [72,73,74].

Specifically, the VLA-4—VCAM1 interaction triggers the NF-κB signaling pathway in MM-MSCs and the transcription and secretion of IL-6, a major MM cell growth factor [5]. In turn, IL-6 enhances the production and secretion of VEGF and basic fibroblast growth factor (bFGF) by MM cells, and both growth factors bind to their receptors on MSCs and re-stimulate IL-6 production [21]. Furthermore, cellular interactions of MSCs and MM cells are mediated through Notch ligands and receptors, and DKK1. The activation of the Notch pathway in MM cells and MSCs induces the secretion of IL-6, VEGF, and insulin-like growth factor (IGF-1) from the latter, which is associated with MM cell proliferation and survival mediated by upregulated expression of survivin [21,75]. On the other hand, DKK1 secreted by MM cells prevents MSCs from differentiating into OBs, and the undifferentiated MSCs can produce IL-6, which in turn stimulates the proliferation of DKK1-secreting MM cells [76,77]. These paracrine loops are critical for maintaining the constant growth of MM cells through the activation of different signaling pathways [78]. The myeloma-MSC interactions, together with the senescent status of MM-MSCs, further enhance the number of cytokines, chemokines, and soluble factors secreted by MSCs to the BM milieu (e.g., TNFα, SDF-1, BAFF, OPN, HGF, IL-10, IL-8, GDF-15, AREG) which may function as MM growth factors and increase MM proliferation and survival [26,27,29,34].

After binding with their receptors, these interleukins and growth factors trigger in tumor cells the activation of signaling pathways such as the JAK2/STAT3, PI3K/AKT, RAS and the downstream mitogen-activated protein kinase (MAPKs), which provoke not only tumor growth and survival but also drug resistance, migration, and dissemination of myeloma cells [72,73].

As previously commented, exosomes from HD- and MM-MSCs were found distinct in terms of miRNA profile (lower content of tumor suppressor miR-15a) and higher levels of oncogenic proteins, cytokines and adhesion molecules (IL-6, CCL2, plakoglobin and fibronectin). Whereas MM-MSCs exosomes increased myeloma proliferation, HD-MSCs were able to significantly reduce MM growth in in vitro and in vivo assays [67]. Similarly, MM-MSCs derived microvesicles were shown to be incorporated by myeloma cells increasing MM viability, proliferation and translation initiation [79]; conversely, HD-MSC-derived microvesicles decreased these effects. Of interest, mass spectrometry analysis revealed that MM-MSCs microvesicles were enriched in VLA-4, which facilitated its uptake and transfer to MM cells [80]. In line with the latter data, MM-MSC-derived exosomes have been shown to mediate the transfer of the lncRNA LINC00461 to MM cells, where it sponged miR15a/16 and enhanced MM cell proliferation and suppression of apoptosis [81]. MiR-10a was also found enriched in MM-MSC EVs, and when conveyed to myeloma cells it enhanced MM proliferation [82]. Concerning MM-derived exosomes, they were found to be enriched in miR-146a and miR-21, which once transferred to MSCs induced the secretion of cytokines and chemokines (including CXCL1, IL6, IL8, IP-10, MCP-1, and CCL5), and increased MSC proliferation and transformation to cancer-associated fibroblasts, finally resulting in increased MM cell viability and migration [55,56] (see Figure 2). 

### 3.2. Contribution of MM-MSCs to Drug Resistance

The presence of surviving tumor cells immediately after therapy (MRD+), suggests the presence of some form of de novo drug resistance. Two types of factors may contribute to this de novo resistance: intrinsic and extrinsic factors. Intrinsic de novo resistance is thought to be caused by pre-existing random genetic mutations that are selected through selective pressures imposed by drugs when these mutations offer a survival advantage (as first shown by Theluria-Delbruck experiment) [83,84,85]. Because of the complexity of acquired resistance, further mutations may be needed in addition to these intrinsic factors to produce highly resistant phenotypes [84]. By contrast, extrinsic factors such as those responsible for environmental-mediated drug resistance (EM-DR) may protect tumor cells that contain intrinsic mutations while other mutations develop. 

EM-DR is rapidly induced by signaling events that are initiated by factors present in the tumor microenvironment and can be subdivided into two categories: soluble factor-mediated drug resistance (SFM-DR), which is induced by cytokines, chemokines, and growth factors secreted by MSCs such as IL-6, IGF-1, IL-1, IL-17 and TNF-α; and cell adhesion-mediated drug resistance (CAM-DR), which is mediated by the adhesion of tumor cell integrins to MSCs or stromal fibroblasts or components of the ECM, such as fibronectin, laminin, hyaluronan and collagen IV [11,86,87,88]. SFM-DR is primarily mediated by the induction of gene transcription, whereas CAM-DR is mediated largely, but not entirely, by non-transcriptional mechanisms including the degradation of activators of apoptosis [89], subcellular redistribution [90], and increased stability of suppressors of apoptosis and cell cycle regulators [91].

Several pieces of evidence have shown that MM-MSCs play an essential role in SFM-DR. The binding of IL-6 produced by MM-MSCs to its receptor on the plasma membrane triggers the activation of MAPK, JAK2/STAT3, and PI3K/AKT pathways, resulting in PC proliferation, survival, and drug resistance [5,72]. Importantly, IL-6 is one of the mediators of dexamethasone resistance in myeloma cells via protein tyrosine phosphatase, nonreceptor type 11 (PTPN11, also known as SHP2) [92,93,94]. IL-6 may also inhibit the antiproliferative effects of cyclin-dependent kinase (CDK) inhibitors p21 and p27 through the PI3K/AKT pathway [95]. Furthermore, IL-6 activation of the JAK/STAT3 pathway induces tumor cell survival by up-regulation/activation of anti-apoptotic proteins MCL-1 and BCL-X_L_, and c-MYC [96]. Clinically, elevated serum IL-6 levels are associated with a poor prognosis and reflect the proliferation fraction of MM cells within patients [97].

Similar to IL-6, IGF-1 produced by MM-MSCs is another important mediator of PC growth, survival, migration, and drug resistance [98]. IGF-1 induced signaling pathways in myeloma cells (PI3K/AKT, MAPK, and NF-κB) that resulted in increased telomerase activity and upregulation of the antiapoptotic molecules, such as surviving cellular FADD-like IL-1β–converting enzyme (FLICE)-inhibitory protein (c-FLIP), X-linked inhibitor of apoptosis protein (XIAP), cellular inhibitor of apoptosis 2 (cIAP-2), and BCL-2–related protein A1 (BFL1) [99,100].

On the other hand, CAM-DR suppresses drug-induced apoptosis through other well-characterized mediators of drug resistance such as P-glycoprotein, fibronectin, laminin, and collagen IV [45]. Research demonstrates that adhesion of MM cells to fibronectin is mediated through integrins such as VLA-4 (α4β1) and VLA-5 (α5β1), among others [86,101]. Fibronectin binding upregulates p27, induces NF-κB activation, and has been shown to alter the expression of 469 gene products in MM cells [102]. Reports also show increased production of osteopontin (OPN) and hyaluronan synthase 1 (Has1) by MM-MSCs [103]. MM adhesion to hyaluronan also confers CAM-DR to MM cells [104], and OPN has been shown to mediate multidrug resistance in other cancers by enhancing hyaluronate binding and may act similarly in MM [105].

In line with the commented data, exosomes from human and murine BM MSCs were found to confer resistance to bortezomib in MM cells [106]. Increased expression of Bcl-2 and full-length caspase-8, caspase-9, caspase 3, and PARP in 5T33MM murine myeloma cells after incubation with BM MSC-derived exosomes was observed [106]. Other studies compared the miRNA profile of circulating exosomes from bortezomib resistant and sensitive myeloma patients. Down-regulation of circulating exosomal miR-16-5p, mi-R15a-5p, miR-20a-5p, and miR-17-5p was found to be predictive for drug resistance to the proteasome inhibitor [107]. Besides, exosomes derived from MSCs from patients resistant to bortezomib (r-MSC exos) conferred resistance to this agent in MM cells, whereas treatment of MM cells with exosomes from sensitive MSCs (s-MSCs) did not [108]. Transcripts PSMA3 and PSMA3-AS1 were enriched in r-MSC exos, and since PSMA3 stability was increased through formation of a PSMA3-AS1 and pre-PSMA3 duplex, the expression subunit α3 of the proteasome was increased conferring acquired resistance to bortezomib (see Figure 2). 

### 3.3. Contribution of MM-MSCs to Dissemination and Homing

MM is characterized by disseminated involvement of the BM, and its progression involves a continuous mobilization of MM cells into the peripheral blood (PB) and homing back to the BM. MM cells are constantly invading new regions within the BM through induced systemic recirculation [109,110]. However, a growing body of evidence has demonstrated a small number of circulating tumor cells (CTCs) in MM and its association with poor prognosis [111]. Current dogma states that tumor PCs depend on the BM microenvironment to survive and expand; nevertheless, as the myeloma progresses, both myeloma cells and the microenvironment become hypoxic, leading to the egress of BM clonal PCs into the PB. Later homing of MM cells to the BM depends on chemokines that regulate the adhesion of MM to MSCs [112].

Mobilization or egress of cells and homing are critically regulated by the CXCL12/CXCR4 axis. CXCL12 (also known as SDF-1α) is produced by MSCs and is the ligand of CXCR4 expressed in PCs [113,114,115]. It has been shown that reduction of CXCL12 or up-regulation of CXCR4 by hypoxia induces the mobilization of PCs out of the BM [116]. Indeed, the BMniche is quite hypoxic (1–2% O_2_) [117]. It has also been shown that hypoxia leads to inactivation of E-cadherin and activation of transcription factors regulating epithelial-mesenchymal transition, including Snail and Twist, indicating that this mechanism can participate in the egress process [118]. 

Regarding homing events, the first step in this process is the MM cell adhesion to endothelial cells (EC) through selectins. The adhesion is mediated by different integrins expressed by MM cells, such as LFA-1 and VLA-4 [119]. This adhesion induces the activation of the CXCL12/CXCR4 pathway which in turn, induces the secretion of IL-6 and VEGF by MSCs, promoting PC proliferation, anti-apoptosis, neo-angiogenesis, and resistance to therapy [67,120]. Furthermore, it has been demonstrated that CXCL12 is highly expressed by MM-MSCs at BM sites of metastatic disease showing a major role in directing homing and trafficking of myeloma PCs [114,115]. Studies to identify the expression of chemokine receptors in MM have shown significant variations in CXCR4 expression ranging from 10 to 100%. CXCL12 secreted by MSCs induces the migration of MM cells in vitro and homing into the BM in vivo. Moreover, CXCR4 blockade led to significant inhibition of migration, homing, and growth, thus halting disease progression [115].

The impact of MSC-derived exosomes in regulating MM progression has been tested in subcutaneously implanted TE-bones using confocal in vivo imaging. Seven weeks after implantation, higher engraftment and tumor burden was observed in mice with TE-bone implants loaded with MM cells treated with MM-MSC exosomes. Notwithstanding, in those implants loaded with MM cells exposed to HD-MSC-derived exosomes, very weak myeloma engraftment was observed [67]. Similarly, MM-MSC-derived exosomes were found to contain cytokines including monocyte chemotactic protein 1 (MCP-1), interferon-inducible protein 10 (IP-10), and SDF-1, and were demonstrated to induce myeloma 5T33MM cell migration in vitro and home to the BM in vivo [106]. In the same line, microvesicles from MM-MSCs conferred increased migration capability to MM cells [79] (see Figure 2).

### 3.4. Contribution of MM-MSCs to Myeloma Bone Disease (MBD): Suppressed Osteoblast and Favoured Adipocyte Differentiation

Osteolytic lesions are a central symptom of MM, and around 80–90% of myeloma patients suffer bone complications at some stage of their disease [121]. Clinical manifestations of myeloma bone disease (MBD) range from diffuse osteopenia to osteoporosis, focal lytic lesions, pathologic fractures, vertebral compression, and vertebral fractures [122]. Not only may MBD greatly compromise the quality of life of MM patients, but most importantly, the presence of pathologic fractures has been associated with decreased survival [123]. This highlights the need for bone-supportive therapeutics complementinganti-myeloma strategies, or ideally, anti-myeloma agents which may also have a beneficial effect on bone [124,125]. 

These osteolytic lesions arise because of disruption of bone homeostasis, with increased bone resorption over new bone formation. Thus, MBD presents a dual component of increased OC differentiation and resorption and, on the other hand, reduced OB formation and bone anabolic activity. During the past two decades, many cellular and molecular mechanisms responsible for the suppressed OB function and increased OC resorption in MM have been identified [126]. Being MSCs the main precursors of OBs, they are the major “passive subjects” of myeloma and other microenvironmental signals leading to impaired OB function; in addition, MM-MSCs also actively contribute to MBD by promoting OC activity and differentiation from myeloid precursors. Finally, we will also comment on the current understanding of MSC fate commitment in the myeloma BM niche, which may favour the adipocyte lineage (see Figure 2).

#### 3.4.1. Mechanisms of Osteoblast (OB) Suppression

In MBD, multiple mechanisms concurrently mediate the impairment of OB differentiation from MSC osteoprogenitors, as well as the diminished bone anabolic activity of mature OBs. OB differentiation from mesenchymal progenitors requires the spatio-temporal integration of Wnt, bone morphogenetic protein (BMP), and Notch signaling pathways, and the transcription factor Runx2/Cbfa1 is considered an essential regulator at the intersection of these pathways leading to OB differentiation [127]. However, multiple intercellular cross-talk of MSCs with myeloma cells or other cells in the BMmilieu (through cell contact, soluble factors, or extracellular vesicles) deregulate the commented pathways or finally affect Runx2 activity, greatly influencing the osteogenic capacity of MSCs. 

For convenience and clarity, mechanisms leading to impairment of osteoblastogenesis and OB function will be presented here as mediated by soluble factors (including inhibitors of Wnt and bone morphogenetic protein (BMP) signaling, cytokines, chemokines, hormones…); adhesion interactions of myeloma cells and MSCs or pre-OBs; altered expression of surface molecules on osteoprogenitor cells and OBs; extracellular vesicles from myeloma cells and miRNAs mediating osteogenic suppression; and DNA and chromatin epigenetic modifications leading to long-term inhibition of OB differentiation.

##### Soluble Factors (Including Inhibitors of Wnt and BMP Signaling; Cytokines, Chemokines, PTHrP)

Extensive in vitro and in vivo studies have demonstrated a crucial role for canonical Wnt signaling in regulating osteoblastogenesis [127,128]. Wnt ligands are secreted glycoproteins which bind to a membrane receptor complex of the low-density lipoprotein receptor-related protein 5/6 (LRP5/6) and Frizzled (FZD); upon binding, dephosphorylated β-catenin is stabilized and translocates to the nucleus where it interacts with the transcription factor T-cell factor/lymphoid enhancer factor (TCF/LEF) to activate the transcription of target genes, such as Runx2 [128]. Elevated levels of two major Wnt signaling inhibitors, such as DKK1 and sclerostin, have been found in serum from MM patients, correlating with the presence of focal bone lesions or the extent of bone disease [129,130]. MM cells are major secretors of DKK-1, which binds to the LRP5/6 co-receptor in MSCs and osteoprogenitors, preventing Wnt binding and thus OB differentiation [131]. Sclerostin also binds the LRP5/6 co-receptor in MSCs and osteoprogenitors, and although it was first found to be produced by osteocytes [132,133], primary and myeloma cell lines also produce this Wnt inhibitor [134]. More recently, Sostdc1 has been found significatively induced when MM and OB lineage cells are co-cultured. This secreted protein serves both as a Wnt and BMP antagonist, leading to suppression of OB differentiation [135]. Other Wnt signaling antagonists, such as the secreted Frizzled related proteins (sFRPs), directly bind to Wnt ligands hindering their effect on OBs [136]. Among the sFRP family members, sFRP2 and sFRP3 are produced by myeloma cells, and inhibit Wnt signaling and OB differentiation. The expression of sFRP3 correlates with clinical bone involvement at diagnosis [137,138].

The BMP pathway has also been recognized as critical in skeletogenesis during development and postnatal OB differentiation and bone homeostasis. The BMPs are members of the transforming growth factor-β (TGF-β) superfamily. After binding to heterodimeric receptors in MSCs and osteoprogenitors, some of them (e.g., BMP2, BMP7) lead to SMAD activation and translocation to the nucleus to directly transactivate Runx2or other osteoblastogenic genes such as *DLX5* (distal-less 5). On the contrary, other BMPR ligands (e.g., Activin A, TGF-β), inhibit OB differentiation [126,139]. MM cells do not secrete Activin A but enhance its secretion by MSCs after their interaction, and OCs also are producers of this factor [140]. Apart from the role of Activin A favoring OC resorption, this factor was shown to inhibit OB differentiation through SMAD2-dependent downregulation of Dlx5 [140]. TGF-β is released from the mineralized bone matrix during bone resorption and has been reported to especially inhibit late OB differentiation [141,142]. Inhibition of OB differentiation by TGF-β is mediated by downregulation of Runx2or Dlx5 expression [143,144]. The HGF is often produced by MM cells and inhibits BMP-induced osteogenic differentiation of MSCs by blocking nuclear translocation of SMADs, thus reducing the expression of Runx2 and Osterix and maintaining progenitors in a proliferative undifferentiated state [145]. In relation to this, transcriptomic profiling of bone lining cells from the 5TGM1 myeloma model revealed BMP signaling to be upregulated in stromal progenitor cells [146]. In vivo treatment with a BMP type 1 (BMPR1a) receptor antagonist or a BMPR1a-Fc-solubilized ligand trap prevented trabecular and cortical bone volume loss by reduction of OC number and reduced OB suppression. However, improved OB mineralization was not achieved when isolated MSCs were directly treated with those BMP inhibitors; rather, the improved OB activity in vivo was related to reduction of Wnt inhibitors DKK-1 and sclerostin. This underscores the reciprocal interaction of BMP and Wnt signaling in MM-MSCs, and preclinical evidence is given for pharmacological BMP inhibition to potentially overcome the uncoupling of bone homeostasis driving MBD. 

Several cytokines and chemokines (i.e., IL7, IL3, or CCL3) have been identified as suppressors of OB function, besides their role favoring OC formation and/or activity [147,148]. IL7 is mainly produced by MM cells and may directly diminish Runx2 transcriptional activity, reinforcing the adhesion-mediated inhibitory effect of myeloma cells on MSCs/pre-OBs [149]. IL7 also indirectly inhibits Runx2 expression through the induction of the Runx2 transcriptional repressor Gfi1 [150]. IL3 is mainly produced by T lymphocytes [151], but also by malignant PCs [147], and has been reported to inhibit basal and BMP2 stimulated OB differentiation indirectly through a CD45+/CD11+ monocyte/macrophage mediator [152]. The pleiotropic CCL3 chemokine, produced by malignant PCs and OCs, was also found to contribute to MBD by osteocalcin downregulation and inhibition of OB function through reduced levels of the transcription factor Osterix [153,154]. Similarly, TNFα inflammatory cytokine has been shown to have OB inhibitory properties by reducing the expression of Runx2 and Osterix [155,156]. In relation to the latter, it is thought that Sequestrosome1/p62 in MSCs and osteoprogenitors mediates the TNFα-induced suppression of OB differentiation in myeloma-MSC co-cultures [157]. More recently, other members of the TNF superfamily, such as TNF-related weak inducer of apoptosis (TWEAK) or LIGHT/TNFSF14, have been shown to inhibit OB differentiation, inducing sclerostin release by OBs [158] or by monocytes [159], thus suggesting new roles and modes of action for sclerostin. In addition, MM cells are known to release the parathyroid hormone-related protein (PTHrP), which binds its receptor in MSCs and OBs, inducing the expression of the transcriptional repressor E4BP4; the latter indirectly inhibits the expression of Runx2 and Osterix through the transcriptional inhibition of cyclooxygenase 2 (COX-2) [160].

##### Adhesion Interactions of Myeloma Cells with MSCs or Pre-OBs 

Cell to cell interactions of human myeloma cells and BMMSCs or pre-OBs were shown to inhibit OB formation and function via blockade of Runx2 activity [147], which was accompanied by a diminished expression of OB differentiation markers (i.e., ALP, osteocalcin, and collagen I). Interactions via VLA-4 on myeloma cells and VCAM1 on MSCs were found partially responsible for this effect since antibodies blocking VLA-4 blunted the inhibitory effect on Runx2 activity [147]. 

The third major signaling pathway implicated in skeletal development regulation, bone remodeling, and MSC differentiation is Notch signaling [125,159]. Notch signaling maintains MSCs in an undifferentiated state at physiological conditions by suppressing OB differentiation [160]. However, in MM, Notch signaling is aberrantly activated [161]. Hyperactivation of Notch ligands in myeloma cells (Jagged1/2) activates Notch transmembrane receptors in MSCs and osteoprogenitors (i.e., Notch 1). Then, the Notch intracellular domain (NICD) is cleaved from the membrane by γ-secretase and translocates to the nucleus to complex the transcription factor CSL and co-activator Mastermind-like (MAML) to initiate transcription of target genes such as *HES* and *HEY*. Hes and Hey inhibit Runx2 activity by direct binding; NICD may also directly interact with Runx2 to repress terminal osteoblastic differentiation [160]. Increased levels of Notch 1 receptor and Notch downstream transcription factors (i.e., Hes1 and Hes5) have been reported in MM-MSCs as compared to HD-MSCs [24], and inhibition of Notch signaling by γ-secretase inhibitors could restore the osteogenic differentiation of MM-MSCs. 

Also, N-cadherin-mediated interactions (CDH2-CDH2) have been found to contribute to myeloma cells’ ability to inhibit osteoblastogenesis [162]. 

##### Deregulated Expression of Surface Molecules such as EphB4 and FZD5/Ror2 on Osteoprogenitor Cells and OBs

Along with the above-mentioned mechanisms, it has been shown that the deregulated expression of cell surface ligands and/or receptors involved in bone homeostasis or osteogenic signaling pathways in MM-MSCs (i.e., EphB4 and Ror2), negatively affects their OB differentiation and function. In this way, the bidirectional signaling between ephrin B2 ligands and EphB4 receptors in the cell surface of OCs and OBs is involved in the physiological maintenance of bone homeostasis balancing OC and OB formation. OCs mainly express ephrinB2, while MSCs and OBs express the receptor EphB4 and ephrinB2 molecules. Forward signaling through EphB4 in MSCs promotes OB differentiation, whereas reverse signaling through ephrinB2 in OCs suppresses OC differentiation [163]. MM cells negatively regulate the expression of ephrinB2 and EphB4 in MM-MSCs as compared to their healthy counterparts. This contributes to uncoupling of bone remodeling, since stimulation of the EphB4 receptor is diminished, leading to reduced osteoblastogenesis, and at the same time, decreased ephrinB2 activation augments OC formation [164]. 

Although the canonical Wnt pathway is central in regulating osteoblastogenesis, the non-canonical Wnt5a ligand has also been shown to promote osteogenic differentiation of human MSCs [165], revealing a regulatory cross-talk between canonical and non-canonical pathways [166]. This non-canonical Wnt signaling is initiated by the binding of Wnt5a to the Ror2/FZD5 receptor [167,168]. MM-MSCs and pre-OBs from myeloma patients exhibited decreased levels of Ror2, and co-culture with myeloma cells downregulated the expression of both FZD5 and Ror2, thus contributing to reduced OB differentiation [167].

##### Extracellular Vesicles from Myeloma Cells and MicroRNAs Mediating Osteogenic Suppression

Recently, EVs from myeloma cells have been shown to target both MSCs/OBs and OCs and thus contribute to MBD [169]. Specifically, exosomes from the murine myeloma cell line 5TGM1 enhanced OC activity and blocked OB differentiation and functionality in both in vitro and in vivo models. Of interest, it was shown that those MM-EVs mediated the transfer of DKK-1, which led to a reduction in Runx2, Osterix, and Collagen 1A1 in OBs [169]. Also, exosomes from MM samples and myeloma cell lines contain the EGFR ligand amphiregulin (AREG), which could be internalized by human MSCs blocking their OB differentiation and promoting the release of the pro-osteoclastogenic cytokine IL8 [170]. But inhibition of OB differentiation through myeloma derived-exosomes has also been mediated by the EV-transfer of lncRNAs and miRNAs. Li and colleagues first demonstrated that MM-exosomes contained lncRNA RUNX2-AS1, which after incorporation by MSCs formed an RNA duplex with RUNX2 pre-mRNA, blocked its splicing, and decreased the expression of Runx2, leading to decreased osteoblastogenesis [171]. Other groups have identified miRNAs in myeloma–EVs with a putative role in bone disease, such as miR-103a-3p, which once transferred to MSCs targeted Runx2 and led to decreased OB formation [172]; or miR-129-5p, which among other transcripts downregulated Sp-1, a transcription factor implicated in osteogenesis, and ALP, a known marker of early osteogenic differentiation [57]. These miRNAs, together with the dysregulated expression of other miRNAs and ncRNAs after myeloma cell contact or soluble factor interaction (e.g., miR-135b, miR-138, miR-221, miR-203a-3p.1, miR-342 and miR-363, miR-223, HOXC-AS3) have been found to lead to the inhibition of OB differentiation from MSC precursors (see comprehensive reviews [43,173]).

##### Long-Term Inhibition of OB Differentiation by Epigenetic Modifications

When looking for candidate transcription factors that could mediate the long-term suppression of OB differentiation, D´Souza et al. found that BM MSCs from myeloma patients and MSCs from MM-bearing mice had increased levels of the transcriptional repressor Gfi1 [150]. Either exposure of naïve MSCs to MM cells or to TNFα and IL7 increased Gfi1 expression and translocation to the nucleus, leading to repression of Runx2 mRNA in MSCs and inhibition of OB differentiation. In fact, after MM exposure, Gfi1 binds the *RUNX2* promoter and recruits several histone-modifying enzymes (i.e., EZH2, HDAC1, and LSD1) that change the bivalent signature of the *RUNX2* promoter into a repressive H3K27me3-prevalent state that blocks its transcription and impedes OB differentiation. Conversely, using inhibitors targeting HDAC1 and EZH2 rescued the expression of Runx2 and enhanced osteogenic differentiation [59,63]. 

In addition, MM-MSCs at different stages of the disease have been found to bear widespread DNA methylation alterations, including Homeobox genes and other genes involved in osteogenic differentiation (e.g., *RUNX2* and *IBSP*), which were associated with differential gene expression [61]. Co-culture of HD-MSCs with myeloma cells partially recapitulated DNA methylation changes of MM-MSCs. In line with these data and those from chromatin-based epigenetic mechanisms, dual targeting of DNA methyltransferases (DNMTs) and the histone methyltransferase G9a promoted the osteogenic differentiation of MM-MSCs in vitro and prevented bone loss in an in vivo model of MM. 

These latter studies underscore the involvement of epigenetic-based mechanisms, both chromatin modification- and DNA methylation-based in the myeloma-induced suppressed osteogenic differentiation of MSCs. Besides, since these epigenetic alterations may be inherited to MSCs’ cellular progeny, they offer at least a partial explanation for the altered transcriptional signature of myeloma MSCs in the absence of myeloma interactions [61,63], and the prolonged suppression of OB differentiation at the site of osteolytic lesions even after remission of active myeloma disease [62].

#### 3.4.2. Mechanisms of Osteoclast Activation

The suppression of OB differentiation in MM renders an excess of MSCs and immature OBs in the BM, which may actively promote OC formation and function through several mechanisms. MM-MSCs and immature OBs show a higher expression of the receptor activator of NF-κB ligand (RANKL) at their membrane, which binds to RANK receptor on OCs and OC precursors, exerting an essential role in the differentiation, activation, and survival of OCs [174,175]. In addition, after interaction with myeloma cells, MM-MSCs secrete less osteoprotegerin (OPG), a soluble decoy receptor for RANKL, thus contributing to the increased RANKL/OPG ratio that promotes OC formation and resorption [176]. As previously mentioned, Activin A is produced by both OCs and BM MSCs after interaction with myeloma cells [140] and has been shown to stimulate OC resorption [177]. Also, increased Wnt5a production by MM-MSCs after interaction with myeloma cells may further contribute to osteoclastogenesis through interaction with FZD/Ror-2 receptors in OC precursors, leading to increased RANK expression and increased sensitivity to RANKL [49,178].

MM cells also produce IL6, soluble RANKL, and many other so-called “OC-activating factors” (e.g., CCL3, IL3, IL7, IL8, IL1β, HGF, CCL20), which have been reported to enhance OC formation from myeloid precursors and OC resorption, and which are out of the scope of this review. OCs, in turn, also release several soluble factors (including IL6, CCL3, OPN, B-cell activating factor (BAFF), and a-proliferation-inducing ligand (APRIL)), which promote MM cell growth and survival, thus creating a vicious cycle between bone lesions and myeloma progression (see reviews [126,179]).

#### 3.4.3. Mechanisms of Adipocyte Formation

The processes of osteogenesis and adipogenesis have traditionally been considered mutually exclusive, since osteogenic or adipogenic differentiation depend on the activation of phenotype-specific transcription factors for spatial and temporal control of gene expression. Similarly, aging and pathological conditions associated with decreased bone loss are inversely correlated with increased BM adiposity [180]. Since both OBs and adipocytes derive from BM mesenchymal stromal progenitors, it is tempting to pose the question that exposure of myeloma cells to MSCs may not only suppress OB differentiation but may also shift differentiation of MSCs towards adipogenesis [181]. Even more, significant plasticity exists between OBs and adipocytes and is the basis for transdifferentiation between the two lineages [182]. Determinants and mechanisms accounting for the favoured adipocyte differentiation from MSCs in the presence of myeloma cells are under study. The onset of adipogenesis seems to be partially dependent on cell-to-cell integrin α4 (myeloma cells)-VCAM1 (MSCs) interactions, which leads to the activation of protein kinase C β1 signaling (PKCβ1) [183]. Later data correlated with increased levels of PPARγ2 in MSCs, which in turn were dependent on the repression of the ubiquitin ligase muscle ring-finger protein-1 (MURF1)-mediated ubiquitination, and thus stabilization of this key adipogenic factor. 

Other factors may as well regulate adipogenesis in the BM. As such, physiologically relevant sclerostin levels, secreted both by osteocytes and MM cells, may induce adipogenesis in human BM-derived MSCs by inhibition of Wnt signaling [184]. MM cells with high heparanase expression have been reported to shift the differentiation potential of OB progenitors to adipogenesis; mechanistically, this shift was due to heparanase-enhanced production of DKK1 by both OB progenitors and myeloma cells [185]. Indeed, DKK1 is secreted by human preadipocytes and promotes adipogenesis [186]. Moreover, chromatin-based epigenetic mechanisms also seem involved in MSC fate reprogramming towards adipogenesis [63] through increased expression of EZH2 and H3K27me3-mediated suppression of Runx2and other Wnt genes, and at the same time permitting the expression of adipogenic factors such as PPARγ and C/EBPα. 

Once differentiated, pre-adipocytes and mature adipocytes seem to play a causative role in the pathology of MM and may affect MBD. BM adipocytes secrete several adipokines and growth factors (e.g., MCP-1, SDF-1α, leptin, TNFα, insulin, resistin) which recruit myeloma cells and promote myeloma growth and protection from chemotherapy [187,188,189]. Adipocytes also secrete adiponectin, which has anti-myeloma properties, and its diminished expression in MM creates a permissive microenvironment for myeloma growth and the development of MBD [190]. Conversely, increased pharmacological levels of circulating adiponectin reduced tumor burden and induced a significant increase in OBs and bone formation rates without affecting OCs [190]. 

### 3.5. Contribution to a Pro-Inflammatory and Immunosuppressive Microenvironment

MSCs are known to display potent immunosuppressive and anti-inflammatory activities, modulating the activity of cells of innate and adaptative immune systems through a plethora of contact-dependent and independent mechanisms [191]. Together with their low immunogenicity [192], these features have granted their use in cellular therapy for the treatment of immunological disorders such as Graft-Versus-Host-Disease [193]. Several groups, however, have shown that MM-MSCs exhibit somewhat impaired immunomodulatory functions. Various studies [26,32,33] reported that MM-MSCs presented diminished immunoinhibitory capability on T cells (reduced inhibition of T cell proliferation, less apoptosis, and less inhibition of T cell activation markers) as compared to HD-MSCs. Besides, when T cells were co-cultured with MM-MSCs, they increased the Th17/Treg ratio compared to HD-MSCs [194]. This altered immunomodulatory capacity was dependent on the altered immunogenicity and secretion profile of MM-MSCs, and, specifically, the increased IL6, VCAM-1, and CD40 were involved in the Th17 population increase [194]. In fact, increased numbers of Th17 cells in MM, along with upregulated levels of IL17 and other pro-inflammatory Th17-associated cytokines have been related to MM growth and progression [195]. Other studies have shown a pro-inflammatory profile of MM-MSCs. Single-cell transcriptomic analysis of the MM BM has identified an activated inflammatory stromal cell population associated with TNF signaling [196]. The increased production of pro-inflammatory cytokines by MM-MSCs after interaction with MM cells has also been reported by several groups [26,34,49]. Since many of these inflammatory cytokines also function as growth factors for myeloma cells or may induce the secretion of myeloma growth factors by other cells in the microenvironment, it is thought that the increased inflammatory microenvironment may also promote MM growth and progression. 

On the other hand, immunosuppression is a common feature of MM patients associated with the evolution of the disease [197,198]. This immunosuppression is mediated by high concentrations of immunosuppressive soluble factors, loss of effective antigen presentation, effector cell dysfunction, and by the recruitment of immunosuppressive populations, such as myeloid-derived suppressor cells (MDSCs) [199,200]. When MSCs were co-cultured with peripheral blood mononuclear cells from normal individuals to generate MSC-educated MDSCs, only MM-MSCs but not HD-MSCs, promoted the induction of granulocyte-like MDSCs with suppressive ability. In the BM microenvironment, stromal cells have been shown to contribute to the expansion and activation of MDSCs through the secretion of hepatocyte growth factor (HGF) and activation of the STAT3 pathway [201]. These MDSCs produced upregulated immunosuppressive factors (e.g., Arginase1 and TNFα) [49] and increased ability to digest bone matrix [202]. Furthermore, these MDSCs have been reported to support MM progression by suppressing T cell responses, inducing Treg differentiation, and even differentiating into OCs (see review [197]). Of interest, MM-MSCs derived exosomes were able to activate MDSCs in the BM and increased their survival by activating STAT3 and STAT1 pathways and increasing BCL-X_L_ and MCL-1 levels. Furthermore, the release of nitric oxide by MDSCs enhanced their immunosuppression on T cells. 

Overall, these pieces of evidence highlight the involvement of MM-MSCs in the modulation of the immune compartment of MM, contributing to an immune suppressive and at the same time pro-inflammatory microenvironment (see Figure 2).

## 4. Concluding Remarks 

In summary, a considerable body of knowledge has been accrued relative to the supporting role of BM MSCs in the growth and progression of MM (contribution to tumor growth and survival, drug resistance, homing and dissemination, myeloma bone disease, and immune suppressive and pro-inflammatory microenvironment). Several underlying changes in MSCs resulting from their interactions with myeloma cells have also been shown to mediate the path along with the transition from normal HD-MSCs to supporting MM-MSCs. These include differential gene expression, non-coding RNA dysregulation and epigenetic (DNA methylation and histone modification) alterations. In fact, the evolution of the disease is paralleled by an evolution of BM MM-MSCs. Special attention has been highlighted on epigenetic modifications on MM-MSCs after cross-talk with myeloma cells, since the epigenetic footprint defines stable phenotypes that can be inherited after cell division and may be responsible for alterations of MSCs in absence of interaction with myeloma cells. Future studies to enlighten the role of MM-MSCs in progression from asymptomatic to symptomatic stages of the disease, dormancy of myeloma cells and relapse are warranted. On the other hand, the possibility of reversal of epigenetic modifications to improve the osteogenic capacity of MSCs or to counter their supportive role in MM, drives attention to epigenetic drugs as potential therapeutic agents in this disease. 

## Figures and Tables

**Figure 1 cancers-13-02542-f001:**
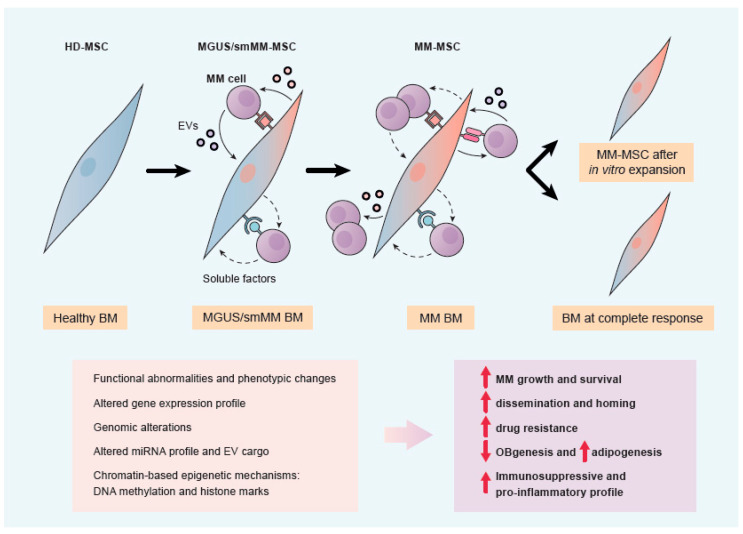
Hypothetical transition from HD-MSCs to myeloma MM-MSCs mediated by interaction with MM cells. Interaction of MGUS and smMM plasma cells with MSCs is considered an initiating event. Direct contact of myeloma cells and MSCs, together with soluble factors (dashed arrows) and EVs (solid arrows) induce various layers of modifications in MSCs (phenotypic, gene expression, genomic, miRNA, and epigenetic), contributing to the transition from HD- to MM-MSCs and to myeloma pathology. Epigenetic modifications may be responsible for maintenance of MM-MSC features in absence of myeloma cell interactions.

**Figure 2 cancers-13-02542-f002:**
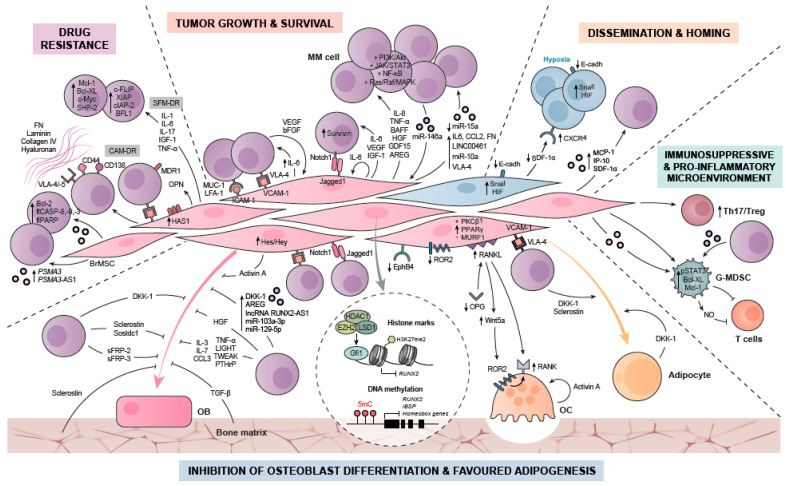
MSC- mediated biological activity in the BM microenvironment of MM. The MSC and myeloma cell cross-talk (through adhesion molecules, via soluble factors, or by extracellular vesicles derived from MM cells or MSCs) actively contributes to the pathology of the disease. The MSC—MM cell interactions increase MM growth and survival, induce drug resistance, promote myeloma dissemination and homing, support myeloma bone disease through impaired OB while favoured adipocyte differentiation and promotion of OC formation, and additionally contribute to a pro-inflammatory and immunosuppressive microenvironment.

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
