# Peer review of "Bone Marrow Mesenchymal Stromal Cells in Multiple Myeloma: Their Role as Active Contributors to Myeloma Progression"

_cancers, 2021, doi:10.3390/cancers13112542_

Round 1
Reviewer 1 Report
This is an excellent manuscript that details the contribution of mesenchymal stromal cells in the pathogenesis of multiple myeloma.
It provides a detailed review on the contribution of MSCs to the growth and survival of myeloma cells, drug resistance, dissemination and homing, myeloma bone disease and the pro-inflammatory state in the marrow.
I have no major suggestions or changes other than minor edits.
On Page 9, section 3.2 should be moved to after section 3.1. Its out of order currently.
On line 215, please correct the acronym of SMM.
This manuscript is an excellent review and will provide a state-of-the-art review on the topic.
Author Response
Response to reviewers
Reviewer #1
This is an excellent manuscript that details the contribution of mesenchymal stromal cells in the pathogenesis of multiple myeloma.
It provides a detailed review on the contribution of MSCs to the growth and survival of myeloma cells, drug resistance, dissemination and homing, myeloma bone disease and the pro-inflammatory state in the marrow.
This manuscript is an excellent review and will provide a state-of-the-art review on the topic.
We are most grateful to the reviewer for his/her comments, and we appreciate the time an effort to review our manuscript. We are most grateful to the reviewer for his/her comments, and we appreciate the time an effort to review our manuscript. As per the reviewer´s suggestions we have rechecked the manuscript for correct spelling, grammar and English style.
I have no major suggestions or changes other than minor edits.
On Page 9, section 3.2 should be moved to after section 3.1. Its out of order currently.
- pages 9-11: we have interchanged sections 3.1 and 3.2 as they effectively were out of order.
On line 215, please correct the acronym of SMM.
- page 8, line 367 and 376: the “smMM” acronym has been corrected.
Reviewer 2 Report
This article is about an important subject.
The introduction needs to be resumed. It is too long.
Ref 15 is not the last update.
l 453 : CD44 is not cited neither in ref 70 nor in 71.
L 883, reformulated : In relation with the latter, other authors have already underscored the epigenetic deregulation that results from the interplay between immune, stromal and cancer cells, and how those epigenetic mechanisms drive tumorigenesis and tumor progression [201].
We dont expect reference in conclusion.
I am not sure that we can say that « Since both DNA methylation status and histone-modifications are potentially reversible, the use of epigenetic drugs targeting myeloma, stromal and immune compartments is emerging as a promising therapy in MM »
Author Response
Response to reviewers
Reviewer #2
This article is about an important subject.
We thank reviewer 2 for his/her time for review and comments to improve the manuscript.
The introduction needs to be resumed. It is too long.
- pages 3-8: We understand that by “Introduction” the reviewer refers to Section 2 of the manuscript: “2. Characterization of Mesenchymal stromal cells in multiple myeloma”, which is the first part of the review. We have tried to resume this section to make it shorter, specially avoiding repetition in Section 3: “Biological roles of MSCs in MM pathology”.
Ref 15 is not the last update.
- page 3, lines 106: we have added two new references (now references 16 and 17) for the latest updates from the International Society of Cellular Therapy regarding the use of mesenchymal stromal cells (MSCs) for clinical purposes.
- new ref 16: Squillaro T, Peluso G, Galderisi U. Clinical Trials With Mesenchymal Stem Cells: An Update. Cell Transplant 2016; 25:829-48. doi: 10.3727/096368915X689622. PMID: 26423725.
- new ref 17: Viswanathan S, Shi Y, Galipeau J, Krampera M, Leblanc K, Martin I, Nolta J,Phinney DG, Sensebe L. Mesenchymal stem versus stromal cells: International Society for Cell & Gene Therapy (ISCT®) Mesenchymal Stromal Cell committee position statement on nomenclature. Cytotherapy 2019; 21:1019-1024. doi: 10.1016/j.jcyt.2019.08.002. PMID: 31526643.
L453: CD44 is not cited neither in ref 70 nor in 71.
- page 9, line 446: we thank the reviewer for the observation about CD44 not being referenced. We have added a new reference (now ref 74) regarding this hyaluronan receptor in myeloma cells.
L883, reformulated: In relation with the latter, other authors have already underscored the epigenetic deregulation that results from the interplay between immune, stromal and cancer cells, and how those epigenetic mechanisms drive tumorigenesis and tumor progression [201]. We dont expect reference in conclusion.
We have finally decided to remove the last paragraph of the “Conclusion section” (page 19, L950-959) to avoid the reference and because it was too hypothetical.
I am not sure that we can say that « Since both DNA methylation status and histone-modifications are potentially reversible, the use of epigenetic drugs targeting myeloma, stromal and immune compartments is emerging as a promising therapy in MM »
We appreciate the reviewer´s comment on this issue. Since the last paragraph of the “Conclusion section” (page 19) has been removed, a new statement has been added at the end of the previous paragraph (page 19, lines 948-949) to underscore the potential of epigenetic drugs as new treatments in MM.
In accordance with the reviewer´s suggestions we have made these changes:
- new ref 74: Neri P, Bahlis NJ. Targeting of adhesion molecules as a therapeutic strategy in multiple myeloma. Curr Cancer Drug Targets 2012; 12:776-96. doi:10.2174/156800912802429337. PMID: 22671924.
Reviewer 3 Report
The authors realized a comprehensive review of the complex role of microenvironment with focus on mesenchymal stromal cells (MSC) in the pathogenesis of multiple myeloma.
The review is well written and organized covering the complex interplay between mesenchymal cells and abnormal plasma cells.
The authors present the different types of interaction between cells including soluble factors, direct intercellular contact via various adhesion molecules as well as extracellular vesicles.
Consequences of exposure of normal mesenchymal cells to multiple myeloma cells are reviewed with focus on intercellular signaling leading to gene expression alterations and epigenetic modifications in MM-MSC. The authors also propose a model of transition of normal MSC to MM-MSC.
The role of MM-MSC in microenvironment changes of multiple myeloma patients is described with detailed presentation of various mechanisms involved in tumor growth and plasma cell survival, dissemination and homing of myeloma cells, immunomodulation of microenvironment, inhibition of osteogenesis and stimulation of adipogenesis and drug resistance. Potential new therapeutic approaches with drugs targeting epigenetic modifications are also mentioned.
In conclusion, the manuscript is a complete and educative presentation of the contribution of MSC to the pathogenesis of MM.
Author Response
Response to Reviewers
Reviewer #3
The authors realized a comprehensive review of the complex role of microenvironment with focus on mesenchymal stromal cells (MSC) in the pathogenesis of multiple myeloma.
The review is well written and organized covering the complex interplay between mesenchymal cells and abnormal plasma cells.
The authors present the different types of interaction between cells including soluble factors, direct intercellular contact via various adhesion molecules as well as extracellular vesicles.
Consequences of exposure of normal mesenchymal cells to multiple myeloma cells are reviewed with focus on intercellular signaling leading to gene expression alterations and epigenetic modifications in MM-MSC. The authors also propose a model of transition of normal MSC to MM-MSC.
The role of MM-MSC in microenvironment changes of multiple myeloma patients is described with detailed presentation of various mechanisms involved in tumor growth and plasma cell survival, dissemination and homing of myeloma cells, immunomodulation of microenvironment, inhibition of osteogenesis and stimulation of adipogenesis and drug resistance. Potential new therapeutic approaches with drugs targeting epigenetic modifications are also mentioned.
In conclusion, the manuscript is a complete and educative presentation of the contribution of MSC to the pathogenesis of MM.
We are thankful to reviewer #3 for his/her effort in reviewing our manuscript and his/her comments. As per the reviewer´s suggestion, we have made corrections for grammar and spelling along the text which are highlighted in red.